# Analysis and Evaluation of COVID-19 Web Applications for Health Professionals: Challenges and Opportunities

**DOI:** 10.3390/healthcare8040466

**Published:** 2020-11-07

**Authors:** Hamid Mukhtar, Hafiz Farooq Ahmad, Muhammad Zahid Khan, Nasim Ullah

**Affiliations:** 1Department of Computer Science, SEECS, National University of Sciences and Technology (NUST), Islamabad 44000, Pakistan; 2Department of Computer Science, College of CIT, Taif University, Taif 21944, Saudi Arabia; 3College of Computer Sciences and Information Technology (CCSIT), King Faisal University, Alahssa 31982, Saudi Arabia; hfahmad@kfu.edu.sa; 4Department of Computer Science & I.T, University of Malakand, Chakdara 18800, Pakistan; mzahidkhan@uom.edu.pk; 5Department of Electrical Engineering, College of Engineering, Taif University, P.O. Box 11099, Taif 21944, Saudi Arabia; nasimullah@tu.edu.sa

**Keywords:** API, dashboard, interface, metadata, visualization

## Abstract

The multidisciplinary nature of the work required for research in the COVID-19 pandemic has created new challenges for health professionals in the battle against the virus. They need to be equipped with novel tools, applications, and resources—that have emerged during the pandemic—to gain access to breakthrough findings; know the latest developments; and to address their specific needs for rapid data acquisition, analysis, evaluation, and reporting. Because of the complex nature of the virus, healthcare systems worldwide are severely impacted as the treatment and the vaccine for COVID-19 disease are not yet discovered. This leads to frequent changes in regulations and policies by governments and international organizations. Our analysis suggests that given the abundance of information sources, finding the most suitable application for analysis, evaluation, or reporting, is one of such challenges. However, health professionals and policy-makers need access to the most relevant, reliable, trusted, and latest information and applications that can be used in their day-to-day tasks of COVID-19 research and analysis. In this article, we present our analysis of various novel and important *web-based applications* that have been specifically developed during the COVID-19 pandemic and that can be used by the health professionals community to help in advancing their analysis and research. These applications comprise search portals and their associated information repositories for literature and clinical trials, data sources, tracking dashboards, and forecasting models. We present a list of the minimally essential online, web-based applications to serve a multitude of purposes, from hundreds of those developed since the beginning of the pandemic. A critical analysis is provided for the selected applications based on 17 features that can be useful for researchers and analysts for their evaluations. These features make up our evaluation framework and have not been used previously for analysis and evaluation. Therefore, knowledge of these applications will not only increase productivity but will also allow us to explore new dimensions for using existing applications with more control, better management, and greater outcome of their research. In addition, the features used in our framework can be applied for future evaluations of similar applications and health professionals can adapt them for evaluation of other applications not covered in this analysis.

## 1. Introduction

The COVID-19 pandemic, caused by the SARS-COV-2 Coronavirus, has affected the human population globally. Due to its higher rate of contagion and its long-term prevalence, it has gained unprecedented attention from the governments, public, and media. Governmental and non-governmental organizations as well as health professionals, scientists, researchers, analysts, statisticians, decision-makers—belonging to any field—are working closely on various aspects of this pandemic. The abundance of information portals, search engines, scholarly databases, and repositories make the right and suitable sources indistinguishable from the irrelevant and incorrect ones. As stated by the World Health Organization (WHO) in an early situation report [1], the rapid global spread of COVID-19 has been accompanied by a “massive infodemic”, i.e., an overabundance of information—some accurate and some not—that makes it hard for people to find trustworthy sources and reliable guidance when they need it. It can cost lives of people as everyone’s safety is at stake and people do not hesitate in believing anything they receive on social media or looking for online solutions to their worries.

However, it is not just the information overload; more alarming is the fact that cybercriminals have found a great opportunity to take advantage of people online [2]. The National Cyber Security Centre (NCSC), UK, has published a warning that cybercriminals are using the coronavirus outbreak as a means of luring Internet users to their sites or to open malicious links via phishing activity https://www.ncsc.gov.uk/news/cyber-experts-step-criminals-exploit-coronavirus). As such, there has been an increase in the registration of webpages relating to the Coronavirus and NCSC has identified a list of over 300 domains that are using the terms "COVID-19" or "coronavirus" in their website address. According to TechRepublic, nearly 2000 malicious COVID-19-themed domains are created every day and more than 86,600 new domains related to the pandemic are considered "risky" or "malicious" (https://www.techrepublic.com/article/nearly-2000-malicious-COVID-19-themed-domains-created-every-day/). According to Clarivate analytics, domain registrations related to the pandemic have surpassed 100,000 since 1 January 2020 and the largest surge occurred in the week following the WHO’s declaration of COVID-19 as a pandemic on 11 March (https://clarivate.com/markmonitor/blog/COVID-19-domains-whats-going-on/). It is also estimated that Coronavirus-themed domains are 50% more likely to be malicious than other domains (https://bit.ly/35fsRcO) (shortened URL). This calls for identifying the legitimate, reliable, useful, and technically correct sources of information that can be readily used by those whose work and decisions have an impact on other people’s lives.

### The Need for COVID-19 Research Applications

The COVID-19 virus was known to the world in late 2019 and after more than 10 months of worldwide efforts, successful treatments of the virus using some vaccine have not been achieved. SARS-CoV-2, the virus responsible for the COVID-19 crisis, is constantly evolving and understanding the trends in its evolution is fundamental to control the pandemic. In decoding the nature of the virus, there have been unprecedented sequencing efforts, so far, producing over 100,000 genomes of severe acute respiratory syndrome coronavirus 2 (SARS-CoV-2) [3]. However, it is not only the SAR-CoV-2 genetic sequencing, the scientists and researchers are trying to figure out the comorbidities, the long-term effects of the illness and subsequent recovery, its transmission and control, among various other aspects on public health. To understand the progress of science against the COVID-19 battle, health professionals and researchers must be aware of the day-to-day development and advancement in COVID-19 research.

The authors faced similar issues when searching up to date and relevant information sources for Coronavirus (or specifically, COVID-19) research. To cater for their own needs as well as to help fellow community members in the research, exploration, analysis, evaluation, and reporting tasks, the authors utilized their information technology skills to identify specialized resources appropriate for the given tasks. As computers and information technology are indispensable part of modern research, particularly in health research [4,5], it was imperative to look for computer applications developed specifically for COVID-19. At the same time, we did not want to explore niche applications suitable for a segment of the population. Thus, instead of searching for desktop or mobile-phone applications, we preferred to identify those applications that are openly accessible from anywhere, by anyone, at any time. Given the ubiquity of the Internet and the Web browsers, we focused on Web-applications for COVID-19. These Web applications (sometimes known as Web apps) are advanced websites with focus on specific functions and involving specific databases to work with. The web-based applications that we have searched and selected are different than and superior to ordinary websites as they have a complex architecture consisting of databases, server-side scripting or programming languages, business or functional logic [6], and information organization that has been specifically created for COVID-19.

Thus, our goal in this work is to help the communities of HP’s working in the capacity of researchers, scientists, and analysts, by providing analysis and comparison of the various online, web-based applications (hereafter referred to as *applications*) that have been specifically developed during the COVID-19 epidemic. Having knowledge of these applications is a need of time for health professionals globally. We believe that knowledge of these applications will enhance their productivity and will allow them to explore new dimensions for using existing applications with more control, better management, and greater outcome of their research. To be more methodological in our approach, we have defined a framework that outlines 17 criteria to be used as features for evaluating the applications.

The rest of the article is organized as follows. We first outline some challenges and the opportunities available to the HP’s to necessitate the importance of knowing these applications. In Section 3, we explain our research methods and our inclusion/exclusion criteria for selecting the indispensable applications for health professionals. Section 4 presents our framework, explaining our features for analysis and evaluation of the applications. Section 5 then presents these applications, detailing their features, identifying their strengths and limitations, as well as discussion on their usage. Section 6 concludes this article.

## 2. Challenges Faced by Health Professionals Due to COVID-19 and Converting Them to Opportunities

As the SAR-CoV-2 is constantly evolving and no confirmed treatment is in sight, the health professionals’ community faces several challenges. Let us first identify some of those challenges followed by how these challenges can lead to new opportunities.

### 2.1. Identifying Most Relevant Research Portals and Resources

As much as there has been an increasing number of infections from Coronavirus, so has been the growth of research publications; analysis tools and applications; dedicated websites and portals; research funding; and the projects dealing with its tracking, analysis, management, cure, and rehabilitation [7]. For people closely monitoring the epidemic, it is imperative to have detailed insights with the latest updates from reliable sources. In the presence of so many information outlets, searching for relevant information sources in various domains of COVID-19 research is one of the methodological challenges in research on COVID-19 [8]. As we discuss later, these information outlets differ in various ways, and choosing the right one for the right job at the right time is not evident.

Recent efforts have resulted in creating various directories and repositories for quick access to selected information, and curated websites dedicated for COVID-19-related information have been created. As of the mid of August 2020, GitHub, the largest open source community for software code and project management has more than 5200 repositories under COVID-19 topic, focused around research and awareness of the virus (https//github.com/topics/COVID-19). The repositories are constantly updated with new lists, packages, data, applications, and programming tools on a daily basis. The repositories are organized by popularity (number of stars given by users on GitHub), number of updates made by the maintainer, or according to the time of recent update. As such curated lists are general purpose and categorized by various themes, they are well suited for general surveying by anyone exploring the pandemic; however, as they are not designed with specific goals in mind, they turn out to be an overwhelming exploration of a sea of information for a researcher who is interested in interdisciplinary work to find the most suitable resource for a given task. As an example, only one of these repositories, the Awesome Coronavirus (https://github.com/soroushchehresa/awesome-coronavirus), lists hundreds of websites categorized into coronavirus information, web and mobile applications, projects, API’s, social media pages and channels, articles, books and hackathons.

### 2.2. Multidisciplinary Research

The nature of the coronavirus pandemic has gathered professionals from multiple disciplines to identify preventive mechanisms [9,10], decode genetic composition of the virus to understand its biological nature [11], invent novel and effective medical treatments [12], control its modes of transmission [13], and devise containment policies in communities or regions [14]. This interdisciplinary endeavor requires having the latest insights from multiple fields spanning across medical, physical, and social sciences as well as multiple fields of engineering. A professional must look beyond her domain of expertise in search of the breakthrough research findings and novel solutions being proposed around the clock. This cannot be achieved by following traditional ways of reading journal articles or news or policy updates only.

### 2.3. Shortage of Area-specific Data and Applications

A biologist, a clinician, an epidemiologist, or an informatician all have different needs and may produce different analyses from the same data about a disease. That is why, as we discuss later, there are hundreds of generic portals providing information about some aspects of coronavirus although the source of data for each of them is the same. This lack of access to specialized information can be seen in various efforts specific to COVID-19. Tableau, the company behind one of the widely-used applications for data visualization, mentions that more than 27,000 visualizations (https://public.tableau.com/en-us/s/COVID-19-viz-gallery) have been published across the world pertaining to COVID-19. However, due to the categorization of these visualizations into broader areas (government, healthcare and public health, economy and business), finding a visualization of interest through their search mechanism is akin to finding a needle in the haystack. The categories are not sorted in any order and the site does not provide any means for searching or filtering through these visualizations. Thus, how can a researcher belonging to a field of research identify the most relevant of these portals that can be useful in his research? That was one of the challenges we faced while analyzing these diverse sources of information: identifying the resources closest to specialized information needs of a health professional or research.

### 2.4. Limitations of Systematic Reviews

A variety of methods can be utilized for the assessment for information sources related to COVID-19, for example, by using only the highly-reputed sources from trusted organizations. In general, systematic reviews are the best source of information assimilation on a given topic of interest. However, in the six-month period from 9 February until 7 August 2020, the NIH office of the portfolio analysis [15] that indexes literature specific to coronavirus disease, had indexed 843 documents containing the term “systematic review” in their title. Going through the titles alone for a subset of these documents is a daunting task and even more difficult is the extraction of relevant information that is not stale. Moreover, production of systematic reviews often takes years [16] and majority of published ones are said to have inaccurate status [17]; so it remains to be seen what is the quality of those rapidly produced systematic reviews [8].

### 2.5. Need for Rapid Synthesis of Evidence

Learning from similar studies, ensuring adequate quality of published research, obtaining and sharing data, are only a few issues pertaining to the lack of appropriate information sources [8]. Due to urgent nature of tackling the epidemic, the traditional approach of waiting for new publications for analysis, evaluation, and knowledge acquisition is a barrier to the advancement of research in the fight against the coronavirus disease. As clinical trials are taking place around the world, health professionals need quick access to the information on successful clinical trials that can be put in practice in similar conditions.

The urgency to address the above-mentioned aspects led to a boom in mostly uncoordinated efforts taken up by individuals, communities, organizations, and governments across the globe [18]. While it may not be evident to overcome all the challenges faced by the health professionals and new challenges may appear frequently, COVID-19 has also created unusual opportunities that may solve some of these challenges and also open undiscovered avenues for research and explorations.

#### Opportunities for Health Professionals in the Era of COVID-19

After an extensive review and analysis of applications developed for COVID-19, we concluded that there is an urgent need to provide the most trusted, up-to-date, and relevant information. As we navigated through this sea of information, we realized new opportunities have emerged, and they keep on rising as new applications are developed and more data on COVID-19 is obtained throughout the globe.

### 2.6. Bringing Positive Reforms

The COVID-19 crisis has resulted in healthcare reforms for longer terms and positive reforms are expected to occur [19]. COVID-19 has emphasized the value of systems thinking [3], implying that small changes can have disproportionate impacts. Programs have been calling on clinicians and consumers to be more conscious of potential harms of unnecessary investigations and treatments [19]. Thanks to system thinking, health professionals have relied more on synthesis of evidence and the involvement of informaticians into application of machine learning and artificial intelligence has given us better results [20,21]. With projections of an overutilized healthcare workforce, oversaturated hospitals, and a shortage of medical support staff, medical students could serve as a vital resource for patient care beyond their traditional roles resulting in early preparation of ready-to-serve professionals. This batch of professionals, having gone through rigorous practical situations, are better equipped than the trainees before them.

### 2.7. Specific Applications for Health Professionals

Health professionals have also learnt that there are great rewards for implementing strong measures early, and for detecting, isolating and contact tracing infected people as rapidly as possible [19], so purpose-built applications have been developed that can be handled by health professionals.

### 2.8. Rapid Understanding of Scientific Progress

The first steps in tackling the epidemic is to understand its nature and the scientific progress in the fight against COVID-19. WHO, CDC, and numerous other organizations have provided basic resources for COVID-19 education. However, as these resources and guides are maintained through human efforts, they may not get updated as quickly as the developments happening around the world. There are some portals, however, that provide automated means for reading summaries and discovering trends in the latest research papers and the conversations around them (e.g., covid19primer.com). By using such resources, professionals are better equipped with the latest developments in COVID-19.

### 2.9. Better Integration for Improved Mitigation

In many cases, the data obtained at the country, state, and city/district levels was found to have poor integration between macro- and micro-levels [19]. Thanks to the visualization dashboards that could depict results in a few clicks on the disparities related to COVID-19 infections, spread, and control, health professionals and policy-makers are now able to understand the dynamics of the disease quickly with the ability to foresee and mitigate its effects.

## 3. Research Methods and Tools

The available literature on COVID-19 is abundant and growing at an exponential scale. For our objective of identifying a variety of COVID-19 research and analysis applications for health professionals, searching through literature would only mean identifying research-based efforts. However, research commercialization and the race of creating better and useful products [22] has resulted in the era of Research and Development (R&D) where startups and innovative firms with focused products are innovating more than the giant players of the domain [23,24]. Thus, in addition to the literature databases for identification of such applications, we also explored open research-based repositories, search engines and social media in search of any developments in COVID-19.

Figure 1 illustrates our methodology. Instead of searching through time-tested and established literary data sources that would search in their general databases of articles, we used search portals specifically developed for tracking COVID-19-related peer-reviewed articles, letters, commentaries, opinions, pre-prints, and case studies. Such portals have emerged as *meta search engines* that combine several databases in a single location. The NIH COVID-19 portfolio (https://icite.od.nih.gov/covid19/search/) is one of such meta search engines that pulls data from PubMed, arXiv, bioRxiv, ChemRxiv, medRxiv, Research Square, and SSRN. Therefore, searching through the NIH portal is preferred than searching through each of the referenced databases as the duplicates are also taken care of. An advantage of searching through specialized portals is that the researcher does not have to include terms like “COVID-19” or “coronavirus”. Using the query operators and filters of the NIH COVID-19 portal (see Section 5.1.2), we mainly searched for articles containing variations on following term combinations: “dashboard”, “visualization”, “analysis tools”, “data exploration”, “question answer”, “natural language processing”, “information retrieval”, “sentiment analysis”, “epidemiological”, “forecasting”, “model”, and “projection”. The search was carried out only on title and abstract of the articles to obtain only those articles that have major portions or contributions related to these terms. The results of these queries were exported as CSV files and then filtered using Python language, removing duplicates, leaving us with 876 articles at the end. The abstracts of these articles were then screened, and 53 articles were retained at the end.

In parallel to the NIH search portal, we searched through Github, the well-known and established code-hosting portal, for COVID-19 topic. Of the 5254 repositories containing the topic, we selected 20 based on the number of stars (more than 1000) and forks (more than 100). Using the inclusion/exclusion criteria (see Section 3), 13 repositories were evaluated and only eight were retained. Finally, we monitored Twitter, Facebook, Research Gate (http://www.researchgate.net), Research Stash (http://www.researchstash.com), and scientific magazines for appearance of any hashtags and titles related to COVID-19 in the month of July 2020. These resulted in 39 distinct web-applications, which were screened for the reputation of developer and inward links pointing to the websites, resulting in 31 candidates. After applying the inclusion/exclusion criteria, we found 21 such applications eligible for our study. Combining the final candidates from these sources, we were left with 58 applications in our final analysis.

Our task was then to analyze each of these 58 applications against 17 features selected by us based upon their functional and technical aspects, and to select the minimum number of applications that can perform the maximum variety of tasks. Therefore, many applications with similar purpose and functionality were compared against one another based on these features and only the *best* one was chosen in the final stage. Our inclusion/exclusion criteria are described briefly in the following subsection.

### Inclusion/Exclusion Criteria

To determine the eligibility of an application in our analysis, we define the following criteria.

For each application, evaluate its usefulness for healthcare researchers, analysts, statisticians, etc. in terms of the 17 features described in Section 4.We categorized applications into three main categories and selected the most common quality features for evaluating each application in a category based on common usability criteria [25] for websites [26].Give preference to those applications which cover broader areas or regions. Country- or state-specific applications were discounted because most of them were similar in nature and those which stand out had either very limited scope of coverage or contained exceptional data collection and processing capabilities and could not be generalized in other cases.Applications intended only for public use were excluded because of their generalization (based on the description provided by the developer and our evaluation of the applications).Criteria like support for multiple human languages or support for multiple types of devices (mobile phones, tablets, and PCs) were not included because English has been the primary language of research around the world and computers are the main devices used by the researchers.In case of identifying many applications with similar purpose or functionality, evaluate each one of them and exclude those which are deficient in features or which have more limitations as compared to those to be included.We excluded all those databases which are maintained by a single publisher (e.g., Elsevier, Lancet, PubMed, etc.) as their repositories are included in the search portals we have included in our list.We further categorized the included applications into different categories based on the nature of the information they provide e.g., portals specific to genetic sequencing and modeling, academic/scientific literature portals, and statistical forecast portals, etc. Competing candidates in the same category were re-evaluated and only the most useful one was retained at the end.

The outcome of our work is available as documented data files in an online GitHub repository (https://github.com/hamidmukhtar/covid-research-tools/) having more details and features compared to as described here and can be downloaded as well as updated through branching requests.

## 4. Framework for Evaluation of COVID-19 Applications

The inclusion/exclusion criteria define the building block of our “framework” for evaluation of COVID-19 applications. The motivation behind the framework is to provide a basic structure to our evaluation at a conceptual level. As mentioned before, the authors only included those applications which are available as Web resources and accessible through a browser (via computer, smartphone, tablets, or similar devices) and did not include mobile apps or desktop applications. Accordingly, in this article, we explore each of them in connection with their respective features, the next layer of our framework. Most of the features are common across all applications, but the supported visualizations and visualization technology features are only used in evaluation of dashboards and the metadata retrieval feature is only used for evaluation of Web portals. Figure 2 depicts these features and in the following we explain them. Table 1, Table 2, Table 3, Table 4 and Table 5 provides detailed analysis of each tool according to these features.

### 4.1. Basic Information about the Application

The basic information about an application consists of several items.

#### 4.1.1. Title/Name of the Application

We identify each application by its title as given by its developer/creator. This title is useful in referencing the application in our discussion.

#### 4.1.2. Aims/Objectives

We describe the objectives of each application as mentioned by its developer, if available, or inferred by our analysis of the application. This is an important distinguishing feature that has helped us in selecting some applications over the others. The objectives of the application will also help the reader quickly identify its suitability for a given task at hand, e.g., literature exploration, visualization of genetic or biological data, question answering, etc.

#### 4.1.3. Web Access Link (URL)

Some applications have dedicated domains specifically created for the disease, some are subdomains of existing organizational domains, yet others are created on personal websites of researchers. Along with other features, our selection of applications has mostly resulted in those URLs which belong to reputed organizations. This is because of the enormous results thanks to the efforts put in by the large teams of researchers and experts compared to the efforts by individuals or small groups.

#### 4.1.4. Regional Coverage and Coverage Period

Both of these features are one of the crucial for selection of applications. We have preferred applications that cover wide areas or regions of the world over those which are narrow in their scope of coverage, e.g., country-, state-, or city-level, with some exceptions to those applications which cover some unique features. As such, almost every country has developed its own dashboard for some aspect of coronavirus tracking. For many countries like Australia, Saudi Arabia, the United Kingdom, or the United States, there are multiple dashboards developed by its ministries, agencies, or research organizations, but we tried to only select the ones that cover indicators at world level. Similarly, applications that support large times (e.g., weeks or months) are preferred over those supporting only daily or hourly statistics.

### 4.2. Interface of the Application

We evaluated the interface of the applications, whether portals or dashboards, for its input and interaction as well as the interface of the results retrieved by the application when a search or selection operation is executed on it.

#### 4.2.1. Interface Type and Complexity

For many search applications, the interface consists of a search box with a search button to start the operation. Some applications support customization of search terms by allowing for ANDing or ORing of the terms. A few applications provide filtering support for specifying the content type on which to apply the search operation. In the case of dashboards, the interface may also support interaction with the visualizations.

#### 4.2.2. Result Data and Refinement

The results returned by a search operation differ amongst the applications. In addition to returning basic information about retrieved articles, some applications provide additional snippets related to search terms. A few applications also support searching within the returned results allowing for better refinement and control over the results.

### 4.3. Input/Output of the Application

Every application processes and possibly modifies some data that is fed into it. The same data can be processed differently by different applications depending upon their objectives. The source and the type of data input by an application are important as much as its output destination. We consider the following features under this theme.

#### 4.3.1. Sources for Data Acquisition

The source from where an application is populated with data is one of the important indicators for its credibility if the developer is not a well-known organization or a government agency. That is why we mention the sources of data acquisition for each application if it is mentioned by the developer or can be assumed. This is also helpful for researchers who might be interested in the original, raw data without any processing applied to it.

#### 4.3.2. Attributes Covered

Data are identified by the types of attributes of data objects. In case of applications designed specifically for COVID-19, these attributes cover the figures, statistics, property, or the characteristics quality of data. We consider these attributes in our evaluation.

#### 4.3.3. Metadata Retrieval

Metadata are the set of data that describes and gives information about other data, in this case data about COVID-19. Considering COVID-19 articles as a type of data, its metadata include the information about the authors, their affiliations, their citations, publisher, etc. Metadata is very useful in identification, sorting, and selection of relevant subsets of data. Applications that support metadata-based filtering offer more customization, flexibility, and powerful operations on obtaining fine results.

#### 4.3.4. Export/Sharing Options

Each application offers its own interface for displaying the modified data from its source. However, sometimes it is more convenient and desirable to convert the output to another form which is better suited for consumption by another process or application. Thus, we report on any facility of exporting the data to some format other than as displayed on the website. An example is exporting a table on a webpage to a comma-separated values (CSV) or Microsoft Excel format or even as an image.

Another aspect is the sharing of data to social media or other websites. This also includes the ability to embed the output as it is in some other website. The advantage of embedding is that any changes at the source are automatically reflected in the embedded data.

#### 4.3.5. Provision of Application Programming Interface (API)

This final feature in the input/output theme of a research application is important where automation is required. An API is a way for computers to exchange information between various applications without human involvement. APIs become essential when one application, process, or tool is dependent on another for its data. Provision of API is one of the major features of success of applications like CORD-19 (Table 1, No. 4), which has thousands of users developing useful analyses and applications on top of it.

### 4.4. Technical Details

#### 4.4.1. Developer

We included the developer of an application as a feature because for many researchers the people behind a cause matter in terms of experience, authenticity, openness, and reputation. While many applications are presumably developed by the “big actors” of the field, we have tried to evaluate every possible application that we come across and, thus, have also short-listed those applications which may be developed by relatively lesser known research group or organization but the application may have some unique or desirable features not found in other applications. For example, the dashboard for Tracking of Emotional Expressions (Table 4, No. 9) has the uniqueness of tracking the emotional expressions of a population, a feature not covered by any other application.

#### 4.4.2. Supported Visualizations

Visualizations are the driving force behind dashboard designs. A variety of dashboards can be developed from the same set of data. That is why we see that some dashboards like Worldometer (Table 4, No. 3) use the same data source to tell a different perspective. In information visualization, it becomes important to see data from different perspectives to gain better understanding and identify salient features or trends in the data.

#### 4.4.3. Visualization Tool or Framework

Visualizations can be created using a certain programming tool or application commonly known as an application or a framework. There are different applications and frameworks for creating visualizations. Given the expertise of developers, analysts, or researchers in specific tools, it is important to know the tool used for developing the visualizations. Most of the researchers in biology, chemistry or medical sciences use R language and we see that the Shiny package in R has been used by many such researchers. On the other hand, informaticians and computer scientists prefer Python or other web-development languages and they have developed visualizations in these technologies.

#### 4.4.4. Update Frequency

For most practical purposes, daily update of statistics is required and sufficient for coronavirus related monitoring. As automation is the key behind the design of most of the applications, the update frequency of the data in an application depends upon the acquisition of data at the data source(s). For some applications, the data collection or update happens at a specific time each day. For others, it is dependent on the availability of data from the bound sources. As the source data may be updated several times a day, some applications update their interfaces as soon as new data is pushed. Because of lack or automation or the necessity for manual intervention, some applications are updated less frequently. We only included the update frequencies for dashboards in Table 4, while those for Web portals are available in the online version in the Github repository.

#### 4.4.5. Licensing

Open source licensing allows the reuse of an application by another developer with a different goal, probably different from the original one or adding another layer of results or analysis on top of it. Therefore, they may be desirable by researchers trying to improve on existing ideas. Proprietary applications may not provide the same level of freedom but may provide some other features not found in the open source applications, e.g., customization or exporting facility.

#### 4.4.6. Support for Customization

Most data exploration tasks can be achieved by changing some data features that may regenerate a graph. For example, adding, removing, or changing the country from the list of countries, selecting a particular date range or the type of variable to observations from the graph options will update the graph accordingly. Such manipulations should be supported by all visualizations to help in the analysis process. When considering multiple visualizations simultaneously, it becomes overwhelming to infer the required information. In such cases, it is highly desirable to be able to customize the layout of the dashboard. The customization feature is not included in this article and can be found in the online version of the data sheet in the Github repository.

### 4.5. Application Usefulness

#### 4.5.1. Target Users

Many applications employ the one-size-fits-all approach and they can be used by the analytics, researchers, decision-makers, and public at large. Such generic applications have been excluded from our analysis. Our objective was to analyze specialized applications targeting specific audiences, particularly health professionals. However, because of the multidisciplinary nature of the required analysis, an application can be targeted for different groups of people. This has been indicated in the applications coverage tables in the article.

#### 4.5.2. Inward Links

It refers to the number of other websites referring to the given dashboard. Generally, a higher number of inward links imply more people talking about it and, thus, more useful the dashboard for the community. For example, the JHU dashboard (Table 4, No. 1) has more than 6 million inward links (backlinks), which is suggestive of its popularity. The continued or rising popularity of a dashboard may be featured to several factors including, update frequency, the data features covered, types of visualizations, etc.

However, the number of inwards links may also be affected by factors such as how early in the pandemic the dashboard was created, which organization/developer is behind the dashboard and who took notice of it when it was created (e.g., a tweet by an influential personality, a notable blogger or a well-known newspaper has a higher and rapid chance of coverage and virality).

#### 4.5.3. Unique Features

If a dashboard stands out from the rest, it is mainly because of its features. Two dashboards offering the same set of observations may be different in terms of other features such as the type of visualization, the ability to customize or export the data. These unique features including the strengths and limitations of each application were included in our analysis but have not been included in this article and can be found in the data sheet in the Github repository.

## 5. COVID-19 Applications for Health Professionals

Considering the variety of resources available for healthcare researchers and analysts, and based on their distinguishing characteristics, we broadly categorize these applications into three main categories:Search portalsVisualization dashboardsEpidemiological models

The search portals category consists of web portals that facilitate searching through one or more literary databases of articles or repositories of scientific information about COVID-19. Table 1, Table 2 and Table 3 present applications under the category of search portals, while Table 4 and Table 5 present the dashboards and epidemiological models together (The statistics reported in the tables were collected in the middle of August 2020 and as they are updated constantly the reader may find different values in later dates.). As it was not possible to accommodate all the columns in a single table, we broke down columns across pages where each application is represented by a number (No.) that is repeated across the tables using the ≻ symbol. The epidemiological models are usually presented as a dashboard of several visualizations so we include them in the dashboards category and they are also described in Table 4 and Table 5.

### 5.1. COVID-19 Search Portals

Every search portal has a distinct objective, and for those with similar objectives and functionality only one of them has been included based on the features chosen in our framework. However, still we can group these search portals into relevant themes based on their goals. We, thus, explain each one under the respective themes as explained below.

#### 5.1.1. Understanding the Scientific Progress

COVID-19 Primer (Table 1, No. 1) describes itself as an application for quickly understanding the scientific progress in the fight against COVID-19. The applications not only indexes the latest publications and research findings but also analyses the emerging topics as the research around coronavirus is evolving. Unlike previously discussed applications that provide search flexibility through filtering, COVID-19 Primer guides the users through its interfaces via navigation menus and lists. This application is a well suited and valuable resource for starter researchers and non-researchers interested in exploring COVID-19 through offered categories, emerging topics, top-cited research, and people, etc. Using this application, users can get a quick overview of where to invest more time for detailed analysis. Researchers with specific intentions and questions may use alternative applications described below.

#### 5.1.2. Searching for Scientific Publications in COVID-19

The applications in this category contain mainly customized or specifically designed search portals and information retrieval systems that search and analyze thousands of documents to extract the relevant information. The types of documents vary from published peer-reviewed articles to preprints, commentaries, case studies, letters, and editorials.

The iSearch COVID-19 application (Table 1, No. 2) can search indexed documents based on simple keywords. Using advanced search options, it gives more control over search operations through additional content type search including types of devices, chemicals, drugs, and conditions. Moreover, using filters, one has greater power over export of results tailored to the requirements. A potential issue with this application is that novice users can have erroneous results if the And/Or criterion is not used correctly. The system is also not robust to special symbols, e.g., "covid-19" and "covid 19" show different results as well as the use of various other special symbols that lead to unpredictable results.

LitCovid (Table 1, No. 3) is also managed by the NIH entity like iSearch (Table 1, No. 2), but unlike the latter, it is more organized around topics. In addition to using open search of articles using keywords, LitCovid provides division of articles under eight topics: General, Mechanism, Transmission, Diagnosis, Treatment, Prevention, Case Report, and Forecasting. By clicking on any topic, a researcher gets a sorted (reverse chronological) list of articles in that topic. The results can be further narrowed down by chemicals, journals, and countries. Another useful feature of LitCovid is the ability to download articles with six different annotations (BioConcepts): gene/protein, drug/chemical, disease, cell type, species, and genomic variants. The interface allows to select/unselect any of these BioConcepts by the researcher and the results will be highlighted accordingly. The annotated articles can be visualized in PubTator [29]. Finally, researchers can create collections of articles using article ID or a search query. This is useful when working on several collections with different characteristics simultaneously.

Covid Scholar (Table 1, No. 4) also provides keyword-based searching of articles. However, the search can be more precise by using phrases inside quotes and some additional symbols or operators, e.g., +(“SARS-COV-2” “coronavirus 2” “novel coronavirus”). For researchers having previous experience of using similar search operators (like those in information technology), this feature can bring flexibility in search but for many researchers, this feature can be prohibiting. One more advanced, and similarly complex, feature is the use of Word Embeddings [31], which is a cluster of words with associated frequencies and that can be visualized. This feature is of interest mainly for exploring connectivity between concepts related to diseases, symptoms, diagnosis, etc.

#### 5.1.3. Question/Answer-Based Information Searching

Applications covered under this category are those which can produce results in response to a question asked by a user in natural language (e.g., English). The Question/Answer (QA)-based systems return superior results compared to simple keyword-based search systems thanks to the recent advances in NLP techniques [32]. This is because by asking a complete question, the system can take advantage of the word-relationships, parts-of-speech, and semantics into account, which is not possible when using keywords only.

The COVID-19 Research Explorer (Table 1, No. 5), by Google, is tailored to “get answers to complex scientific questions related to COVID-19”. Instead of searching through a global database of web documents, as done by the Google search engine, the Research Explorer returns answers from the publications specific to COVID-19 only. Thus, answers returned are scientifically valid statements, claims, or experimental results. This application uses data from the CORD-19 (Table 1, No. 10) system.

Another application that uses QA for information retrieval is covidAsk (Table 1, No. 6) [32]. It is designed on top of BioBERT-based language models [33,34] and incorporates an entity-level search engine called BEST (Biomedical Entity Search Tool) [27] to display important entities relevant to the question. Because it uses only the abstract of articles obtained from CORD-19 (Table 1, No. 10), our comparison of covidAsk and COVID-19 Research Explorer finds the latter to be performing better in QA tasks.

#### 5.1.4. Searching for Genetic and Biological Data

KnetMiner (Table 1, No. 7) is an integrated data platform for gene mining and biological knowledge discovery [35], for searching and visualization of connected data to explain complex traits and diseases [36] and for supporting evidence-based gene discovery and complex trait analysis across species [37]. Its database can be searched with keywords, gene lists, or genomic regions. It was designed before the coronavirus pandemic but now it has COVID-19 specific knowledge network whose detailed breakdown of 19 types can be found on their website (https://knetminer.org/COVID-19/html/release.html).

The Lens Human Coronaviruses Data Initiative (Table 1, No. 8) is built with the objective of extracting knowledge that is otherwise hidden in patents, scholarly works, and biological sequences. It is a suite of applications each one accessible through its own app interface from the main website. Each application specializes in a specific search function providing several filtering options to narrow down the search to specific authors, journals, conferences, countries, date range, subject matter, etc. A useful feature of Lens is provision of APIs for patent sequencing or scholarly articles so users can build customized services on top of Lens.

The COVID-19 Data Portal (Table 1, No. 9), maintained by the European Bioinformatics Institute (EBI) offers interfaces for searching through viral sequences, host sequences, expressions, proteins, biochemistry resources and other literature. Every type of resource can be searched through its own interface; however, the search mechanism is rudimentary offering a single text-field for entering keywords. The results can be interpreted in detail by going through the section-wise breakdown of items according to various topics. As the wider European COVID-19 Data Platform is behind this portal, it has a larger user base and facilitates researchers across the world to upload their data. One of the limitations of the portal is its capability to export only raw data and has a limited set of analysis tools for researchers. A similar application, Nextstrain SARS-CoV-2 for genome analysis has been included in the dashboards (Table 4, No. 7).

#### 5.1.5. CORD-19-Based Applications

CORD-19 (Table 1, No. 10) is one of the biggest endeavors in the list of all search-related applications. It is a suite of research applications that is built around a growing dataset, the biggest among all the COVID-19 datasets. All applications are accessible from the main interface of the website. The CORD-19 dataset can be downloaded but its potential lies in various online accompanying applications.

The *Semantic Scholar* (https://www.semanticscholar.org/cord19/get-started) application is a keyword-based search engine of articles. The *Adaptive Research Feed* (https://www.semanticscholar.org/feed/create?name=COVID-19) application uses Artificial Intelligence to generate custom feeds for researchers based upon their initial interactions. Once the researcher’s intent is learnt, the application provides up to date information in the form of feed to the researcher.

The *SPIKE-CORD* (https://spike.COVID-19.apps.allenai.org/) application can be used to generate powerful extractive queries with support for Boolean expressions, Tokens, and Structured queries that can employ several search operators. With its extended capabilities, researchers can narrow down their search results to only a few articles or a single article that can match the exact intent of the researcher.

The *SciSight* (https://scisight.apps.allenai.org/) application is for exploring the evolving network of science in COVID-19. It can visualize the association between different concepts, based upon the literature. One can use it to find what groups are working in which directions, see how biomedical concepts interact and evolve over time, and discover new connections.

The *SciFact* (https://scifact.apps.allenai.org/) application is designed for scientific claim verification in articles. Using advanced Natural Language Processing (NLP) techniques, the application can assess whether a scientific paper provides evidence supporting or refuting a scientific claim. The claim can be asked by the researcher as a sentence in English language.

### 5.2. COVID-19 Tracking Dashboards

A dashboard is a reporting tool designed for tracking and visualization of data. We have classified dashboards into different categories as follows.

#### 5.2.1. Disease Tracking, Statistics, and Analysis Dashboards

Almost every country is maintaining an information dashboard providing different tracking, statistics, and analysis reports, mainly targeted for the public. The states or provinces of some countries have their own dashboards that provide even more detailed analysis than the national level. Our objective is to cover only those dashboards which cover global data on COVID-19 as that would satisfy the needs of most researchers. For researchers working on data for a specific state or country may, it may be useful to access the relevant dashboard. For example, the official COVID-19 dashboard by the government of Pakistan (http://covid.gov.pk) has more details than the dashboards covering global cases.

One of the earliest dashboards for Coronavirus tracking [41] was developed by Center for Systems Science and Engineering (CSSE) at Johns Hopkins University (JHU). Known as the JHU Coronavirus Resource Center (Table 4, No. 1), it is a complete portal for a variety of COVID-19 resources. The main reason behind including JHU Resource Center in our list of dashboards is that JHU provides the collected data freely available in its GitHub repository (https://github.com/CSSEGISandData/COVID-19), which is starred by around 24,000 users, showing its usefulness, and forked about 15,000 times, which determines its reuse by other users. Furthermore, the article [41] describing the dashboard has been cited over 1500 times and the application has more than 6 million backlinks by other websites. These statistics about the dashboard make it the most useful resource for COVID-19 data and visualization. Its main tracking dashboard provides real-time updates of coronavirus cases across the world. The dashboard provides different visualizations: an interactive global map showing the cumulative cases, the number of global cases, a sorted list of cases by country, global and country-wise death and recovery cases, and a graph displaying the trend of daily cases. The testing section provides some visualizations for testing patients in US states only, and the tracing section has information and guidelines relating to contact tracing for COVID-19.

The Data Explorer by Our World in Data (Table 4, No. 2) [42] provides drilled-down details of Coronavirus statistics. This is a useful application for public but it also provides detailed analyses and in-depth reports on policy responses, outbreak monitoring by countries, international and domestic travel restrictions, tests, progress in vaccines, mortality risks, the indirect impact of the pandemic and several other guides about the prevention and risk factors for the coronavirus disease. Due to its wide coverage of the pandemic and its associated factors, we found it to be a useful resource for policymakers, data scientists, statisticians, human sciences, and social sciences researchers.

The Worldometer Coronavirus Pandemic dashboard (Table 4, No. 3). Its distinguishing features include the main table displaying global data for 16 different features related to COVID-19. The table can be viewed as a list of global countries or as continent-wise breakdown of countries. The features include total cases, deaths, recoveries, serious and critical cases, total tests as well as various ratios per million of the population. Historical detailed data of each country is available as charts. The website also provides demographic conditions of the cases, incubation periods, symptoms, transmissions, and the WHO risk assessments.

The New York Times Coronavirus Map (Table 4, No. 4) has some unique data visualizations not offered by the other dashboards. The hot spots map and the color-coded weekly cases provide a quick view of the epidemic development in each country during the past 7 days. A 14-day historical data is provided separately for cases in countries where they are increasing, decreasing or remain the same. It also provides maps of special cases such as deaths above normal, school reopening, lockdown status, treatments, and vaccines development and testing tracker.

#### 5.2.2. Clinical-Trials Tracking

The Global Coronavirus COVID-19 Clinical Trial Tracker (Table 4, No. 5) is useful for understanding of the trials taking place in the international effort to tackle the COVID-19 pandemic. Its main global map provides an interactive mechanism to click on highlighted areas to see the details of trials/sites, binding, patient settings, enrollment range, treatments, and outcomes. The treatment networks can also be visualized. The data is also displayed in a tabular form and can be downloaded for offline, custom analysis.

We chose this clinical trial tracker over a few others, including the WHO clinical trial tracker, as it uses data from several sources, has interactive visualization covering many features related to trials, and has about 1000 backlinks by other websites.

#### 5.2.3. Disease Modeling Dashboards

The Nextstrain SARS-CoV-2 portal for genome analysis (Table 4, No. 7) provides a suite of data visualization applications for pathogen sequence data analysis [38]. These applications incorporate SARS-CoV-2 genomes as soon as they are shared and provide analyses and situation reports. A genetic application maps how deadly viruses, including SARS-CoV-2, spread around the world and gives an overview of viral mutations and what they mean (and do not mean) for the COVID-19 pandemic.

#### 5.2.4. Sentiment and Mental Condition Tracking Dashboards

As much as the coronavirus disease has effects on people’s physical health, so has its effects been evaluated on mental conditions [43,44] and the emotional states [45] of people. Because large-scale tracking of the mental and emotional states of people cannot be done through physical devices, some research work has focused on use of social media and text analysis for evaluating such conditions specific to the coronavirus pandemic [46]. Such works become more useful when presented as visualizations that can be used by policymakers and social science researchers for evaluating the evolving mental state of people to predict various demographics, economic, and lifestyle changes.

The Penn COVID-19 US Twitter Map [39] (Table 4, No. 8) analyzes the personal information from tweets of US citizens to automatically identify the chronological and geographical distribution of COVID-19 cases in the United States. The dashboard also has a graph depicting the change in anxiety, sentiment, and loneliness of people. Other graphs depict the common topics in healthcare and panic buying due to COVID-19, top symptoms mentioned on Twitter and COVID-19 tweets per capita.

Pellert et al. [40] developed a dashboard (Table 4, No. 9) for tracking of emotional expressions of population in Austria. The authors retrieve data from the news platform derstandard.at, Twitter, and a chat platform for students to highlight changes of language use during COVID-19 in comparison to a neutral baseline. They constructed special word clouds to visualize that overall difference. Using time series analysis, the determine the changes or spikes in anxiety, anger, social terms, positive emotions, as well as sadness over time in the society.

### 5.3. Epidemiological Models

Epidemiological models attempt to forecast the pandemic days and months ahead and take action to change the course of the pandemic for the better. These model are either mathematical [47] or statistical [48], use numerical modeling and computational simulation [49], or apply the state-of-the-art machine learning techniques for short-term [50] or long-term [51] projections of the disease. The particular model mainly depends upon the type of data available to the researchers. These models are frequently updated due to change in the parameters and data. To address the forecasting of the coronavirus pandemic, we came across hundreds of research articles and dozens of dashboards, each one dealing with a different set of parameters for modeling the problem or creating projections. However, because of their constant need for update, very few models are maintained for a longer period and their usefulness vanishes over time.

The IHME COVID-19 Projections (Table 4, No. 6) aim to help policy-makers plan for the pandemic. The dashboards produce projections for total and daily deaths, daily infections and testing, hospital resource use, and social distancing across the globe. Depending upon the data, it can also model the projections for state or province levels. To help policy-makers understand how different policy decisions could affect the trajectory of the pandemic in their location, they developed three different scenarios. The current projections are based on current mandates (closing of educational facilities, non-essential businesses, ban on large gatherings, and mandatory mask usage), the mandates easing scenario projections are based on changes in the mandates, and the universal mask scenario assumes 95% mask usage with the current mandates. Thus, the projections can be adapted for different scenarios.

### 5.4. Discussion on COVID-19 Applications

The present analysis is helpful in identifying the contribution of major players in COVID-19 analysis and research. It can be seen that the CORD-19 (Table 1, No. 10) dataset is used by several other applications, albeit slightly differently due to the way they process and display the data. All these applications return results on some user query, but the flexibility in search, filtering, and result analysis differ thanks to the advancement in NLP techniques and their applications in COVID-19 research, e.g., drugs recommendations [52], semantic textual similarity of COVID-19 texts [53], tracking positive CT imaging features of respiratory illness [54], and so on. As such, a researcher may like to use these applications in parallel and analyze the results collectively. With experimentation, and with the capability to understand the advanced search and filtering mechanism of each application, researchers may be able to identify the best application for analysis of data, publications, genetic encodings, etc.

We reported on a handful of applications among thousands present online. Although the authors worked with assiduity to define the selection features and used the criteria carefully for inclusion or exclusion of various applications, considering the prolonged prevalence of the pandemic it is quite probable that new applications with better functionality or added benefits appear in the near future. The authors have made available the collected data for general benefit and use in a Github repository that can be used as a guideline by anyone interested in evaluating a new application. The data sheet (uploaded as a comma-separated values file) also contains additional details with some features not included in this article by virtue of the brevity of presentation. Interested readers may be directed to consult the repository. We deliberately chose generic features applicable to a broad range of applications. New features can be added to address specific classes of applications or address their peculiar aspects. It is desirable that an automated framework be developed that can use machine learning and natural language processing to automate the evaluation.

## 6. Conclusions

Due to the complex nature of the COVID-19 epidemic and its mechanism of spread and transmission in the global population, a one-of-a-kind and all-purpose approach for its treatment may not be applicable in COVID-19 research. Health professionals are required to embrace a multidisciplinary approach in addressing the challenges raised by the pandemic. In this article, we described a criteria-based framework consisting of 17 features for evaluation of the applications needed by the health professionals in their fight against the COVID-19 pandemic. The variety of applications we consider for evaluation cover different stages of research and analysis such as understanding the scientific progress; analyzing latest research articles; exploring the emerging topics and trends; visualizing the relation between symptoms, drugs, and pathogens through semantic word clouds; and obtaining and analyzing the latest decoded genetic sequences related to the SARS-COV-2 virus. Our framework, currently consisting of generic features, can be extended with area-specific guidelines and features. In the future, the authors intend to apply machine learning and natural language processing techniques that can automate the evaluation process to a large extent. An automated application will be developed that can be used by independent researchers to evaluate any application using custom features. 

## Figures and Tables

**Figure 1 healthcare-08-00466-f001:**
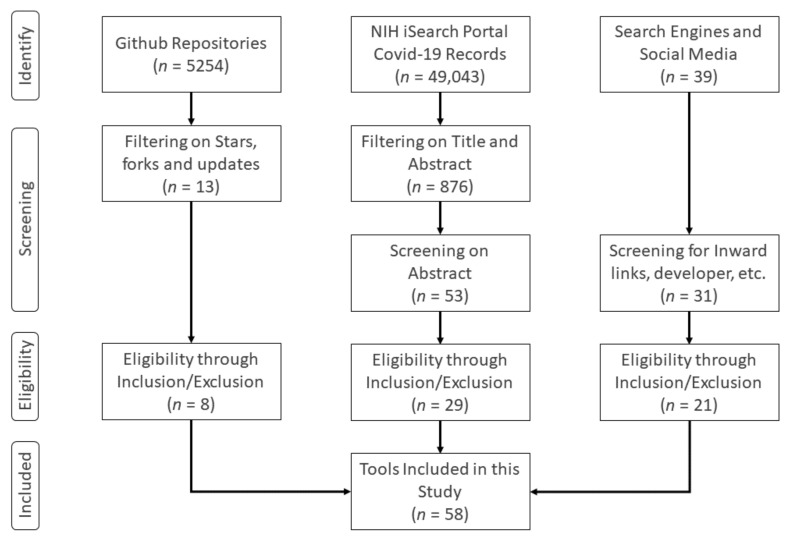
Diagram of systematic identification, screening, eligibility, and inclusion/exclusion of applications for selection in this study.

**Figure 2 healthcare-08-00466-f002:**
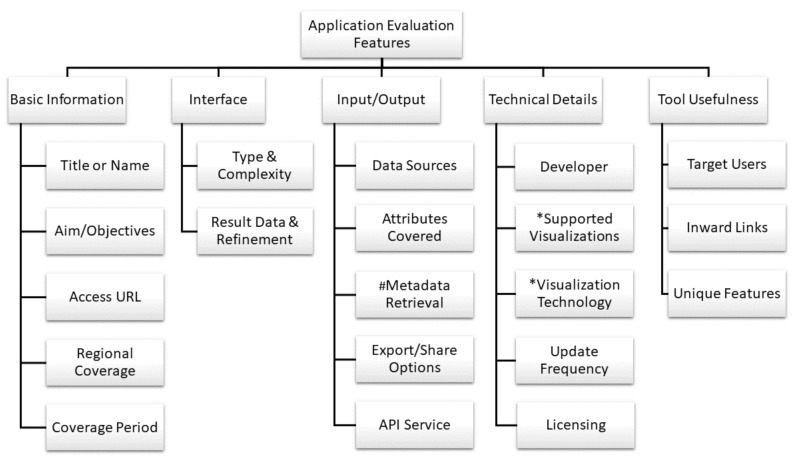
Selected features for evaluation of Web applications: portals, dashboards, and epidemiological models (* applies only to dashboards, # does not apply to dashboards and epidemiological models).

**Table 1 healthcare-08-00466-t001:** Analysis of COVID-19 search portals using selected features (Part 1 of 3).

No.	Name/Title	Objectives/Goal	Access	Data Sources
1	COVID-19 Primer	Quickly understand the scientific progress for COVID-19	https://covid19primer.com/dashboard	PubMed, bioRxiv, medRxiv, arXiv
2	iSearch COVID-19 Portfolio	Searching for Peer-reviewed, preprint articles, letters to the editors, and journal comments	https://icite.od.nih.gov/covid19/search/	Articles from PubMed and preprints from arXiv, bioRxiv, ChemRxiv, medRxiv, Research Square, and SSRN
3	LitCovid	Curated literature hub for tracking up-to-date scientific information about COVID-19	https://www.ncbi.nlm.nih.gov/research/coronavirus/	PubMed
4	Covid Scholar	Using Artificial Intelligence (Natural Language Processing) to power search on research papers related to COVID-19	https://covidscholar.org/	CORD-19 Dataset (Semantic Scholar), Elsevier Novel Coronavirus Information Center, LitCovid, The Lens
5	COVID-19 Research Explorer	Get answers to complex scientific questions related to COVID-19	https://covid19-research-explorer.appspot.com/	CORD-19 (Semantic Scholar)
6	covidAsk	Questions, Answers in real-time	https://covidask.korea.ac.kr/	CORD-19, Allen Institute for AI papers
7	KnetMiner	Ranking of genes and visualization of integrated biological data within an easy-to-use, web environment, in a graph format	https://knetminer.org/COVID-19/	KnetMiner proprietary
8	The Lens Human Coronaviruses Data Initiative	Scholarly research works metadata and biological sequences from patents in a machine-readable/explorable form	https://about.lens.org/COVID-19/	Microsoft Academic, PubMed, CrossRef, Core, and WIPO
9	COVID-19 Data Portal	Facilitates data sharing and analysis in order to accelerate coronavirus research	https://www.covid19dataportal.org/	ENA, UniProt, PDBe, EMDB, Expression Atlas and Europe PMC and lit. sources
10	CORD-19 (COVID-19 Open Research Dataset) and Related applications (See Section 5.1.5)	A group of applications and resources on COVID-19 to bring together computing community, biomedical experts, and policymakers to fight COVID-19 [20]	http://www.semanticscholar.org/cord19	Web crawling, WHO, Google Scholar, Microsoft Academic Search, CiteSeerX, etc.

**Table 2 healthcare-08-00466-t002:** Analysis of COVID-19 search portals using selected features (Part 2 of 3).

No.	Intended Users	Complexity of Query	Results Returned	Result Refinement
≻ 1	Researchers, policy-makers, public	Keyword and phrase-based query can be made	List of publications with title, publication date, journal name, and article source	Results can be refined according to certain criteria such as Date, last 7 days, etc.
≻ 2	Researchers in need of general COVID-19 literature by trusted sources, with support of various metadata search	Support searching through the metadata fields (title, abstract, journal, author, devices, conditions, chemicals, and drugs)	List of publications with title, publication date (sorted), journal name and link to details and PDF document	Does not support searching within the result
≻ 3	Scientific literature into the biology of the virus and the diagnosis and management of those who have been infected	Each article automatically annotated with six different entity types Gene, Disease, Chemical, Mutation, Species, CellLine with color coding through PubTator Central	Abstract and link to the journal for full article as well as annotated six different entity types for further selection of the entity	Results can be refined for various domains such as chemical And/Or journals, etc. One can search within the related articles
≻ 4	Researchers in COVID-19 and related domain	The search can be narrowed down by selecting various options such as peer-reviewed articles or not	List of publications with title, publication date (sorted), journal name and link to source of the article	Select an article for details of the meta data as well as other related articles through document embedding
≻ 5	Researchers, scientists	Question is formulated in natural language (English)	List of publications with title, highlighted possible answers, journal name and link to the article	Results can be refined by follow-up questions into original query
≻ 6	Researchers	Question can be formulated in natural language English	Answers and displays important entities relevant to the questions from BEST [27] and CORD [20]	Only can copy the title from answer section and can navigate in the entities section
≻ 7	Biomedical, genetic, and clinical research	Explore genes related to the search inputs, according to networks of connected knowledge [28]	Accession, gene name, CHRO, start, evidence	A number of views can be selected from Gene View, Map View, Evidence View, Network View
≻ 8	Medical and biological Researchers	Query can be made very complex over range of parameters with Query Editor	List of patent/publications with title, authors name, publication date, journal name,	If any one result is selected, then it gives details of the articles metadata as well.
≻ 9	Virologists, genetics researchers, biochemists	Search through keywords only	A list of items with some description, specific to type of search	The results are divided into subsections according to subject or topic and can be explored further
≻ 10	Global research community	Supports different queries depending upon the sub-tool/application (see in discussions)	Semantic Scholar: list of publications with title, publication date (sorted), journal name and link to source of the article	Results can be refined for the conference or journal, date, publication type, and authors

**Table 3 healthcare-08-00466-t003:** Analysis of COVID-19 search portals using selected features (Part 3 of 3).

No.	Metadata Retrieval	Export/Sharing of Results	Unique Features	API Service	Latest Data
≻ 1	In the sub interface for papers, paper title, authors’ names, journal name, DOI	Not available	Combines dashboards, papers, and social media at one place	None	48,031 papers
≻ 2	Journal information, article content (abstract, title, condition, chemicals and drugs, target, devices), authors’ data, bibliometrics, etc.	Results with customized features selection described in the metadata	Curated manually by experts, offers various statistics related to articles, sources,	None	52,907
≻ 3	Overall lists more than 30 features to choose from	Annotated publications in batches, in BioC, PubTator or JSON formats. RSS feeds for specific topic	Curated by ML and then manually, Categorized by different research topics and geographic locations for improved access	PubTator Central (PTC) API [29]	36,239 publications
≻ 4	Journal information, article content (title, abstract, authors’ names), tag (treatment), etc.	Depends if publisher provides support	Combines patent data for COVID-19	None	67,000 COVID-19 specific articles
≻ 5	Title of the relevant paper, Journal name, date of publication	Not available	Use of neural network model for terms retrieval	None	More than 50,000 articles according to Google
≻ 6	Title of the relevant paper, PMID	Not available	Uses NLP to process papers and answers question using NER process	http://github.com/dmis-lab/covidAsk	Data is from CORD-19 database
≻ 7	Gene specific data such as ACCESSION, GENE NAME, CHRO, START, EVIDENCE	Sharing with registered collaborators supported	Graph patterns and search ranking techniques for genes relevance to search words	Available as KnetSpace API	genes: 27,599, concepts: 674,969, relations: 1,652,520
≻ 8	Abstract, Access information, affiliations, citing patents, citing works, etc.	Citation can be exported as RIS, BibTex, CSV, JSON	This search portal contains patents for COVID-19 as well scholarly research work	PatSeq Bulk Download and scholarly API are available	Not available
≻ 9	Depending on the search type additional details (e.g., organism, gene, length, strain, taxonomy) are shown	Data can be downloaded in bulk	Data can be submitted by users; part of the wider European COVID-19 Data Platform	None	52,271 viral sequences, 890 host sequence, 62 expressions, 582 proteins, 1463 biochemistry records
≻ 10	Journal information, article content (title, abstract, authors’ names)	Not available	Blend of ML, NLP, and MV for adding semantic analysis and ontology into the Semantic Scholar [30]	Semantic Scholar API: http://api.semanticscholar.org/	186,000 articles mentioning “COVID-19” indexed

**Table 4 healthcare-08-00466-t004:** Analysis of COVID-19 dashboards and epidemiological models using selected features (Part 1 of 2).

No.	Dashboard Name/Title	Link	CoverageSince	Update Frequency	Data Sources
1	JHU Coronavirus Resource Center	https://coronavirus.jhu.edu/	2020-01-01	Daily, Real-time	Curated by CSSE team
2	Our World in Data: COVID-19	https://ourworldindata.org/coronavirus	2020-01-01	Daily	European Center for Disease Prevention and Control (ECDC)
3	Worldometer: COVID-19 CORONAVIRUS PANDEMIC	https://www.worldometers.info/coronavirus/	2020-01-01	Daily, Real-time	Official websites, social media accounts, press briefings and daily report released by health ministries, government institutions, government authorities
4	The Net York Times Coronavirus Map: Tracking the Global Outbreak	https://www.nytimes.com/interactive/2020/world/coronavirus-maps.html	2020-01-01	Daily	Local governments, CSSE/JHU and WHO
5	Global Coronavirus COVID-19 Clinical Trial Tracker	https://www.covid-trials.org/	2020-05-01	Weekly	WHO International Clinical Trials Registry Platform, various countries’ Clinical Trials Registries, clinicaltrials.gov
6	IHME COVID-19 Projections	https://covid19.healthdata.org/	2020-03-30	Depending on data	Local and national governments, hospital networks and associations, the World Health Organization, third-party aggregators
7	Genomic epidemiology of SAR-CoV-2	https://nextstrain.org/sars-cov-2	2019-12-01	Daily	Various Research Groups around the world,
8	Penn COVID-19 US Twitter Map	https://www.arcgis.com/apps/opsdashboard/index.html	2020-05-08	Daily	Tweets collected from Twitter API, CSSE/JHU
9	Tracking of Emotional Expressions	http://www.mpellert.at/covid19_monitor_austria/	2020-04-22	Daily	derstandard.at, Twitter

**Table 5 healthcare-08-00466-t005:** Analysis of COVID-19 dashboards and epidemiological models using selected features (Part 2 of 2).

No.	Open Source	Technical Details	Visualization Technology	Regional Coverage	Export/Share Options	Backlinks	Target Users
≻ 1	No	On website Data and code also available	Plotly	Global	Facebook, Twitter, LinkedIn, Email	6,150,102	Public, data scientists, medical science researchers
≻ 2	Yes	On website Data and code also available	Open Graph Protocol, Polyfill IO	Global	CSV	270,243	Policy-makers, data scientists, statisticians, human and social science researchers
≻ 3	No	On website Data also available	ArcGIS, Open Graph Protocol	Global	Twitter, Facebook, Email	8,943,983	Public, statisticians, researchers
≻ 4	Yes	On website	Open Graph Protocol, CSS, JavaScript	Global	Facebook, Twitter, LinkedIn, Email	143,564	Public, journalists, researchers
≻ 5	No	On website Data also available	Shiny, Leaflet	Global	CSV	994	Researchers, policy-makers
≻ 6	No	On website	Javascript/SVG	Global		245,631	Policy-makers
≻ 7	Yes	On websiteResearch article [38] Data and code also available	HTML/Javascript	Global	Not available	4778	Genetics researchers, virologists, epidemiologists, public health officials, and community scientists
≻ 8	No	Research article [39]	ArcGIS	USA	Not available	Not available	Public, researchers, policy-makers
≻ 9	No	Research article [40]	Flexdashboard, Plotly	Austria	Not available	7	Public and policy-makers

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
