# Peer review of "Analysis and Evaluation of COVID-19 Web Applications for Health Professionals: Challenges and Opportunities"

_healthcare, 2020, doi:10.3390/healthcare8040466_

Round 1
Reviewer 1 Report
The topic of the paper is very interesting with a lot of information provided.
Some point of improvements are as follows,
- topic is interesting but you don’t actually introduce the problem you address and don’t formulate a problem statement or business questions based on this problem statement.
-abstract line 9 and 11 what do you mean by TASK here? I can see some missing information flow.
-there is a very big confusion in the use of words such as TOOLS. I can see in the paper scientific literature, tools used and platforms and methods all of them discussed in a very mixed manner. See figure 1, you started with three sources and output is tools(n-53), why you merged repositories, search engines, and NIH portals?
-Straucture of the paper needs a major revision, I will recommend the structure in section4 with some modification
-
- Search engines
- repositories
- Visualization dashboards
- Epidemiological models
Section 4 talks about said division but it should be visible from the beginning.
what is the relationship between the parameter and the division provided in section4? you should explicitly say which parameters are useful for which type of studies such as search engine use parameters x, y, and z, and for models we use parameters a, b, and c. Also mention why, motivate your choice. Motivation against certain choices is not clear at all.
Figure2: many parameters are discussed but in the medical field, we have a large amount of data, the scalability of the tool should be included in these parameters. the caption says that these attributes are used for search engine and dashboards. are these not used for models? If not then why?
Sometimes these are called attributes, sometimes 17 parameters and sometimes a framework. Be consistent with what you want to say and how you want to say.
The caption of all tables talks about selected parameters? what are there selected parameters? Are these the attributes mentioned in section in section 3?
Section 4 talks about the dashboard but what about Epidemiological models, you should discuss them as well.
section 4.3 talks about NLP techniques. have you tried to see NLP techniques if yes then name then instead of saying that there are different NLP techniques.
Reviewer 2 Report
In the world of politicized information (and misinformation) and cyberthreats, this research is timely. My only reservations are two-fold: 1) The use of English language needs some polishing. 2 the authors sugget that they have “described a criteria-based framework consisting of 17 parameters for evaluation of research and analysis tools needed by the health professionals in their fight against the Covid-19 pandemic.” However, the use of the term “evaluation of research and analysis tools” can be misleading as much of what is described is tools and not research.
Round 2
Reviewer 1 Report
Check Spelling and apply english grammer correctly.
